DISCOVERY REPORT

# The *Mycobacterium tuberculosis* PE15/PPE20 complex transports calcium across the outer membrane

**Vishant Boradia**[1], **Andrew Frando**[1], **Christoph Grundner**[1,2,3] *

**1** Center for Global Infectious Disease Research, Seattle Children's Research Institute, Seattle, Washington, United States of America, **2** Department of Pediatrics, University of Washington, Seattle, Washington, United States of America, **3** Department of Global Health, University of Washington, Seattle, Washington, United States of America

* christoph.grundner@seattlechildrens.org

The Editors encourage authors to publish research updates to this article type. Please follow the link in the citation below to view any related articles.

**Data Availability Statement:** All relevant data except RNA seq are within the paper and its

## Abstract

The mechanisms by which nutrients traverse the *Mycobacterium tuberculosis* (*Mtb*) outer membrane remain mostly unknown and, in the absence of classical porins, likely involve specialized transport systems. Calcium ions ($Ca^{2+}$) are an important nutrient and serve as a second messenger in eukaryotes, but whether bacteria have similar $Ca^{2+}$ signaling systems is not well understood. To understand the basis for $Ca^{2+}$ transport and signaling in *Mtb*, we determined *Mtb's* transcriptional response to $Ca^{2+}$. Overall, only few genes changed expression, suggesting a limited role of $Ca^{2+}$ as a transcriptional regulator. However, 2 of the most strongly down-regulated genes were the *pe15* and *ppe20* genes that code for members of a large family of proteins that localize to the outer membrane and comprise many intrinsically disordered proteins. PE15 and PPE20 formed a complex and PPE20 directly bound $Ca^{2+}$. $Ca^{2+}$-associated phenotypes such as increased ATP consumption and biofilm formation were reversed in a *pe15/ppe20* knockout (KO) strain, suggesting a direct role in $Ca^{2+}$ homeostasis. To test whether the PE15/PPE20 complex has a role in $Ca^{2+}$ transport across the outer membrane, we created a fluorescence resonance energy transfer (FRET)-based $Ca^{2+}$ reporter strain. A *pe15/ppe20* KO in the FRET background showed a specific and selective loss of $Ca^{2+}$ influx that was dependent on the presence of an intact outer cell wall. These data show that PE15/PPE20 form a $Ca^{2+}$-binding protein complex that selectively imports $Ca^{2+}$, show a distinct transport function for an intrinsically disordered protein, and support the emerging idea of a general family-wide role of PE/PPE proteins as idiosyncratic transporters across the outer membrane.

## Introduction

Second messengers are a class of signaling molecules that permit a fast response and swift amplification of signals intracellularly. Among second messengers, calcium ions ($Ca^{2+}$) are a particularly versatile signal in eukaryotes [1]. The functions of hundreds of human proteins are directly regulated by $Ca^{2+}$, and almost every cellular process is affected by $Ca^{2+}$ [1,2]. $Ca^{2+}$

Supporting Information files. All RNA seq files are
available from NCBI-GEO (accession no.
GSE214266).

**Funding:** This research received funding from
Division of Intramural Research, National Institute
of Allergy and Infectious Diseases, Grant numbers:
R01AI117023, R01AI158159, R21AI137571 to CG,
Grant number: 5T32AI053396 to AF. The funders
had no role in study design, data collection and
analysis, decision to publish, or preparation of the
manuscript.

**Competing interests:** The authors have declared
that no competing interests exist.

**Abbreviations:** $Ca^{2+}$, calcium ion; CBP, calcium-
binding protein; CM, membrane; CTSM, Chelex-
treated Sauton's medium; CW, cell wall; CYT,
cytosolic; DE, differential expression; FRET,
fluorescence resonance energy transfer; ICP-OES,
inductively coupled plasma optical emission
spectrometry; KO, knockout; PDIM, phthiocerol
dimycocerosate; SAM, significance of microarray;
WT, wild type.

signaling is initially facilitated by $Ca^{2+}$ flux along membrane $Ca^{2+}$ gradients that are carefully maintained by pumps and transporters.

Several bacteria also maintain a similar $Ca^{2+}$ gradient between the inside and outside of the cell [3]. What's more, $Ca^{2+}$ has anecdotally been linked to bacterial processes such as motility, spore formation, gene expression [4,5], and, in the case of *Yersinia*, also to virulence [6], suggesting a common bacterial $Ca^{2+}$ sense-and-response system. However, while some components of $Ca^{2+}$ signaling have been identified in bacteria, their number remains small. $Ca^{2+}$ transporters have been annotated but few experimentally tested. Similarly, $Ca^{2+}$-binding proteins have been predicted, but few experimentally confirmed. As a result, $Ca^{2+}$ remains a poorly understood signaling mechanism in bacteria, and many fundamental questions about $Ca^{2+}$ homeostasis remain unanswered, from the triggers of $Ca^{2+}$ influx, the $Ca^{2+}$-binding proteins, to the eventual cellular outcomes.

*Mycobacterium tuberculosis (Mtb)* is surrounded by a highly impermeable outer cell wall that is composed primarily of the complex phthiocerol dimycocerosates (PDIMs) that form an outer membrane [7], creating a structure not unlike that of the outer membrane of gram-negative bacteria. While transport through the outer membrane in gram-negative bacteria is facilitated by characteristic beta barrel porins [8], no equivalent porins have been identified in *Mtb*. The impermeable outer membrane and lack of porins raise the question how *Mtb* transports small molecules such as nutrients, metabolites, but also $Ca^{2+}$ across the outer membrane. In this way, the very first step in *Mtb* $Ca^{2+}$ signaling remains unknown.

The mycobacterial PE/PPE proteins have long been a mystery. They are predominantly found in pathogenic mycobacteria, where they take up a substantial share of the coding capacity [9], and many are substrates of a type VII secretion system [10,11]. The PE/PPE proteins are associated with the outer membrane of the mycobacterial cell wall [12,13]. Many PE/PPE proteins have been implicated in aspects of tuberculosis pathogenesis, but the molecular mechanisms have not been conclusively identified [14]. A recent milestone study showed that several PE/PPE protein pairs function as channels for nutrient transport across the outer mycobacterial membrane [15], defining an idiosyncratic transport system and suggesting a new and perhaps shared family-wide function for the PE/PPE proteins as small molecule transporters.

Here, we sought to further explore $Ca^{2+}$-mediated processes in *Mycobacterium tuberculosis*. We identified regulation of ATP levels and biofilm formation by $Ca^{2+}$. The transcriptional response to $Ca^{2+}$ was narrow, and the most highly regulated genes were *pe15* and *ppe20*. We show that PE15/PPE20 form a complex, directly bind $Ca^{2+}$, and facilitate the influx of $Ca^{2+}$ into *Mtb* across the outer membrane. These data point to a functional $Ca^{2+}$ sense-and-response system, identify physiologic processes regulated by $Ca^{2+}$ and identify a new, specific PE/PPE $Ca^{2+}$ import system across the outer membrane.

## Results

### $Ca^{2+}$ affects ATP levels and biofilm formation

The role of $Ca^{2+}$ in *Mtb* physiology is almost entirely unknown. We initially tested for parallels with $Ca^{2+}$ effects on other bacteria. $Ca^{2+}$ transport in *Escherichia coli* depends on ATP, and $Ca^{2+}$ in turn increases intracellular ATP levels [16]. To test if $Ca^{2+}$ has a similar effect on ATP levels in *Mtb*, we exposed *Mtb* to increasing concentrations of extracellular $Ca^{2+}$ and measured the intracellular ATP levels. ATP levels increased by >2.5-fold in the presence of 1 mM $Ca^{2+}$ and by >4-fold in the presence of 10 mM $Ca^{2+}$ (Fig 1A). These changes are larger than those observed in *E. coli*, where 10 mM $Ca^{2+}$ resulted in 30% elevation in ATP [16]. To test the reverse effect, we cultured *Mtb* in the presence of EGTA, a $Ca^{2+}$-specific chelator and observed a reduction in ATP levels by approximately 50% (Fig 1A).

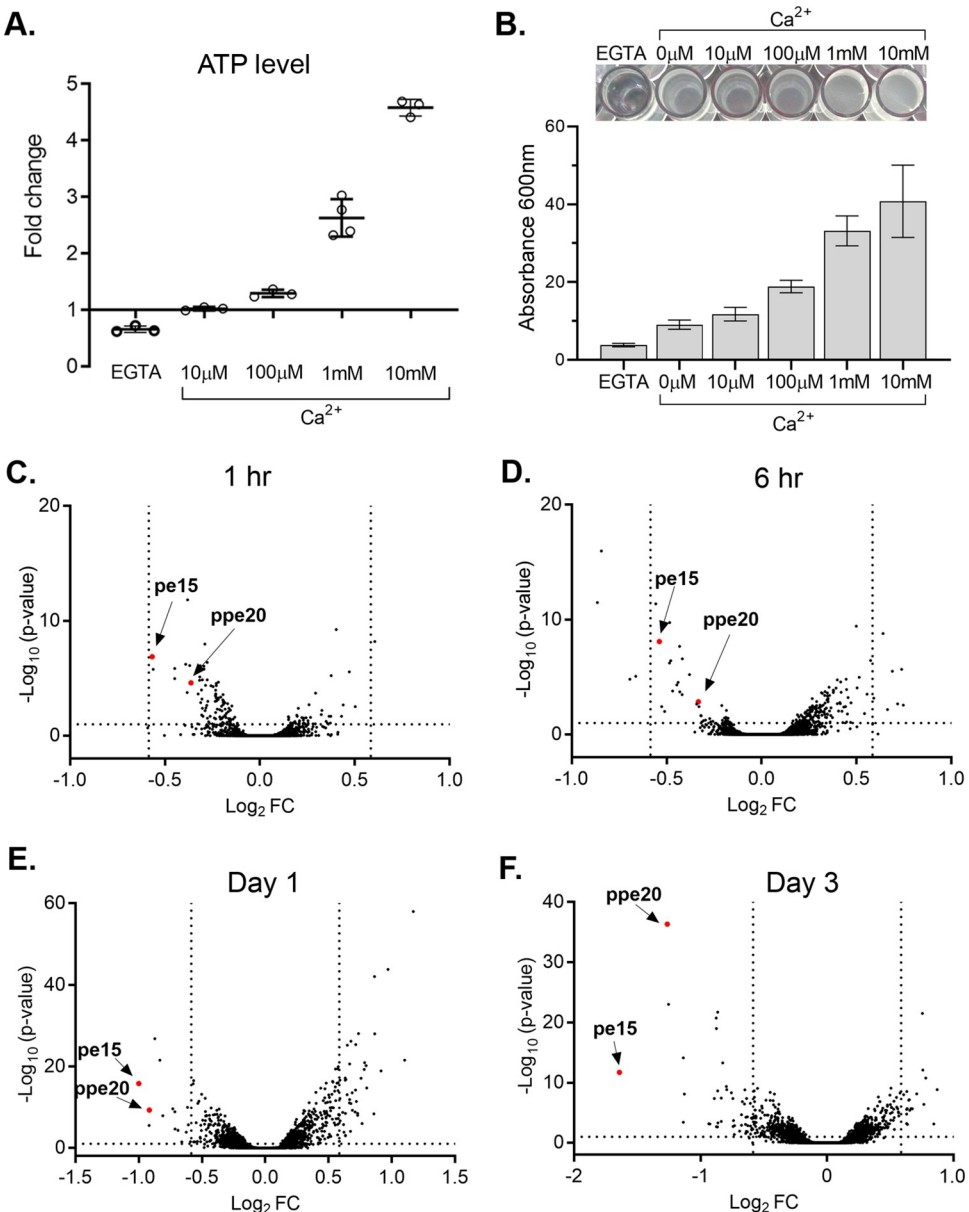

**Fig 1. Ca²⁺ affects *Mtb* cellular processes and down-regulates *pe15* and *ppe20* transcripts.** (A) Effect of Ca²⁺ on cellular ATP levels. *Mtb* was grown in detergent-free CTSM containing increasing concentration of Ca²⁺ (10 μm–10 mM) or 1 mM EGTA at 37°C, treated with 0.1% Tween-80 and ATP was quantified using BacTiter-Glo reagent. Fold change was calculated by comparing to the Ca²⁺-free condition. Data are from biological triplicates, error bars show standard deviation. (B) Ca²⁺ promotes biofilm formation. Biofilms were grown in the presence of increasing concentrations of Ca²⁺ (10 μm–10 mM) and 1 mM EGTA and were quantified using crystal violet. The experiment was repeated 3 times and a picture of a representative experiment is shown. Error bars show standard error. RNA-seq analysis of *Mtb* transcripts after exposure to 1 mM Ca²⁺ for (C) 1 h, (D) 6 h, (E) 1 day, (F) 3 days. Significant down-regulation of *pe15* and *ppe20* transcripts occurred at 1 and 3 days. Few other genes are significantly changed. The data underlying all the plots in this figure are included in S1 Data. Ca²⁺, calcium ion; CTSM, Chelex-treated Sauton's medium.

In some bacteria, a link between biofilm formation and Ca²⁺ has been proposed [17,18]. To test this idea in *Mtb*, we measured the generation of biofilm in response to increasing concentrations of Ca²⁺ in vitro. Indeed, 1 mM Ca²⁺ led to a 4-fold increase in biofilm when compared

to low $Ca^{2+}$ conditions (Fig 1B). To rule out that this increase is simply a result of increased growth and thus biomass in high $Ca^{2+}$ conditions, we tested for differences in growth of *Mtb* in the same $Ca^{2+}$ concentrations as above. We detected no differences in growth (S1A and S1B Fig), showing that the biofilm effects are specifically due to biofilm generation, not to differences in biomass.

### *pe15*/*ppe20* transcripts are down-regulated in response to $Ca^{2+}$

In eukaryotes and in some bacteria, $Ca^{2+}$ affects the transcription of a large number of genes [19]. To test for transcriptional effects of $Ca^{2+}$ in *Mtb* and to identify genes potentially involved in $Ca^{2+}$ homeostasis, we grew *Mtb* with and without 1 mM $Ca^{2+}$ and determined transcriptional effects by RNA-seq. At early time points, few transcripts changed abundance in response to $Ca^{2+}$, although we detected a larger response after 3 days of $Ca^{2+}$ exposure. All changes in transcript abundance are given in Table A in S1 Table. Interestingly, 2 genes with reduced transcript abundance were apparent as early as day 1 and were the most down-regulated genes by day 3: *pe15* (Rv1386) and *ppe20* (Rv1387) (Fig 1C–1F). This behavior in response to $Ca^{2+}$ was reminiscent of metal transporter genes that are often regulated in response to changing metal concentrations, with importers typically down-regulated and exporters typically up-regulated in high metal concentrations.

### PE15/PPE20 form a complex, bind $Ca^{2+}$, and localize to the cell wall

Several PE/PPE proteins have been shown to form protein complexes, typically those coding in the same operon [20]. The *pe15* and *ppe20* genes are also co-operonic, suggesting that they could be a functional protein pair. According to previously published data, recombinant expression of PE15 and PPE20 individually in *E. coli* failed to produce soluble protein [20]. However, we could readily obtain soluble, recombinant protein when the 2 were expressed together, also indicating a potential interaction. To conclusively show a PE15 and PPE20 interaction, we tested for an association by reciprocal pulldowns with tagged proteins. For all biochemical experiments, we co-expressed His-tagged PE15 and Strep II-tagged PPE20 from a dual expression plasmid and precipitated separately with beads binding to each tag. Both proteins were efficiently pulled down by both beads, indicating binding between PE15 and PPE20 (Fig 2A). To further test the interaction, we co-expressed both proteins and visualized them by native PAGE. Both proteins migrated together, as shown by imaging for the respective tags, further confirming that they form a complex (Fig 2B).

To test whether PE15 and/or PPE20 directly bind $Ca^{2+}$, we incubated recombinantly expressed PE15/PPE20 with $Ca^{2+}$ and tested for protein stability using a thermal shift assay with a gel readout (Fig 2C). After heating and precipitation, PPE20 showed clear differential stability when incubated with $Ca^{2+}$, indicating $Ca^{2+}$ binding. By using different concentrations of $Ca^{2+}$ in the same assay, we determined a $Ca^{2+}$ denaturation curve and estimated a $K_D$ of 383.7 ± 34.08 μm (Fig 2D and 2E). Interestingly, in a departure from typical ligand-induced stability changes, $Ca^{2+}$ binding decreased thermal stability of PPE20. PE15 did not produce interpretable results in this assay. Ligand binding in the context of transport suggests that the PE15/PPE20 complex may be a specific channel, as channels typically bind to their ligands [21]. To confirm $Ca^{2+}$ binding of PE15/PPE20 in an orthogonal assay, we used equilibrium dialysis of $Ca^{2+}$ in the presence and absence of the PE15/PPE20 complex (Fig 2F). After complete equilibration, the fluid chamber containing PE15/PPE20 retained 30% more $Ca^{2+}$ than the fluid chamber without protein as determined by inductively coupled plasma optical emission spectrometry (ICP-OES) (Fig 2G). Controls showed a similar effect of a known $Ca^{2+}$-binding protein and no effect for a protein that does not bind $Ca^{2+}$ (Figs 2G and S2).

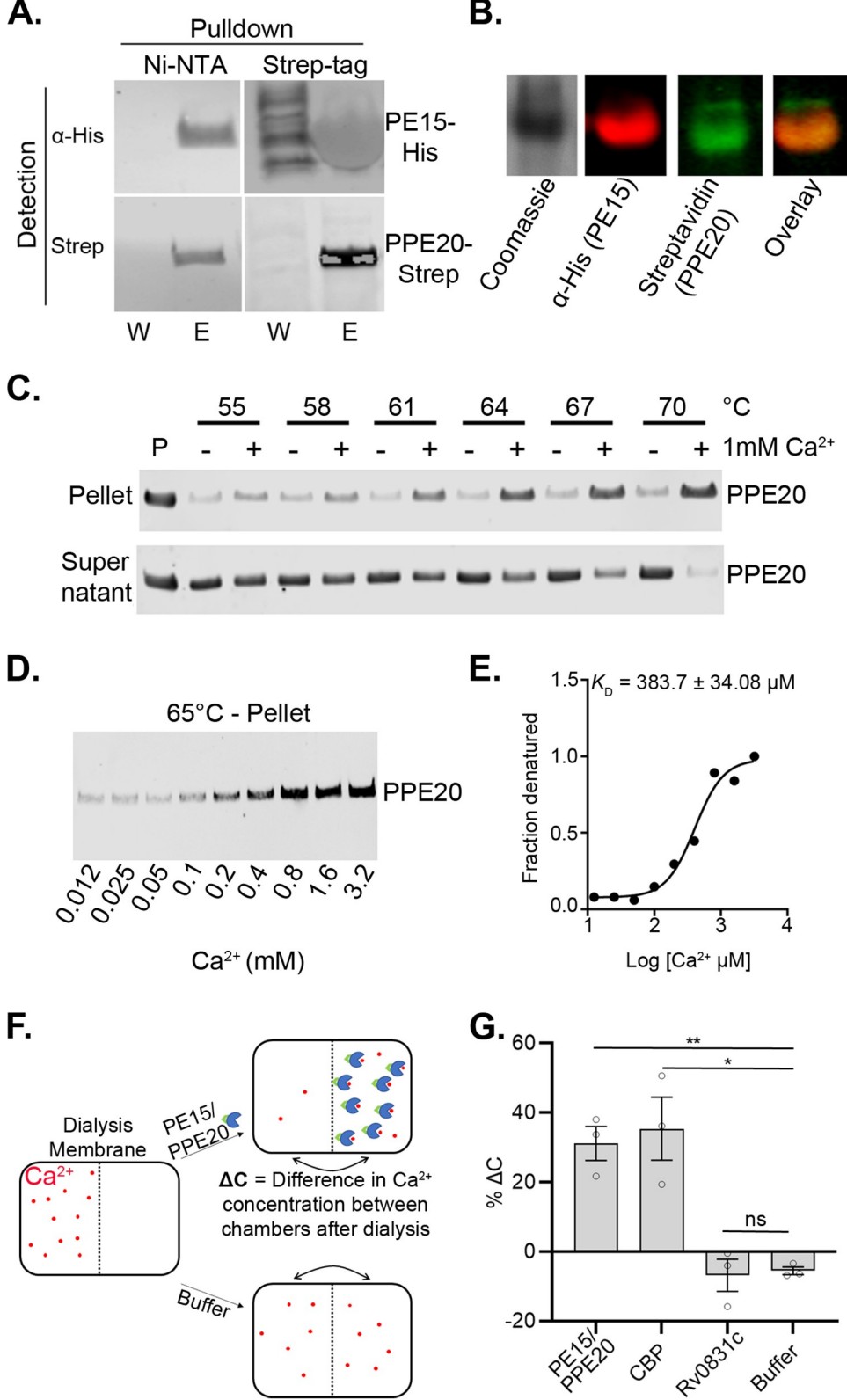

**Fig 2. PE15 and PPE20 form a complex and PPE20 binds Ca²⁺.** (A) Western blot of PE15 with an N-terminal His tag and PPE20 with a C-terminal Strep tag shows the 2 co-purify. W: wash, E: elution. (B) PE15 and PPE20 co-migrate on a native PAGE gel, also indicating complex formation. (C) Thermal shift experiment with western blot readout

shows different thermal stability of PPE20 in the absence and presence of $Ca^{2+}$. The PE15/PPE20 complex was used, but PE15 did not show interpretable difference. (D) PPE20 stability is $Ca^{2+}$ dose dependent. (E) Band intensities in (D) plotted to estimate the $K_D$ of PPE20 for $Ca^{2+}$ (383 μm). (F) Schematic of equilibrium dialysis experiment. (G) The PE15/PPE20 complex and a known $Ca^{2+}$-binding protein (CBP) show $Ca^{2+}$ retention in the protein chambers. Rv0831, a protein not known to bind $Ca^{2+}$, does not show $Ca^{2+}$ retention. A total of 10 μm of protein was used and $Ca^{2+}$ measured by ICP-OES. The data underlying all the plots and uncropped images in this figure are included in S1 Data and S1 Raw images. $Ca^{2+}$, calcium ion; CBP, calcium-binding protein; ICP-OES, inductively coupled plasma optical emission spectrometry.

To test if PE15/PPE20 localize to the outer membrane as do other PE/PPE pairs [15,22], we tested for PPE20 expression in different *Mtb* cell fractions. PPE20 was only detected in the cell wall fraction (Fig 3A).

## PE15/PPE20 KO reverses $Ca^{2+}$-dependent phenotypes

To test for a phenotypic link of the PE15/PPE20 protein complex in $Ca^{2+}$-dependent processes, we generated an *Mtb pe15/ppe20* knockout (KO) strain by recombineering [23]. We next

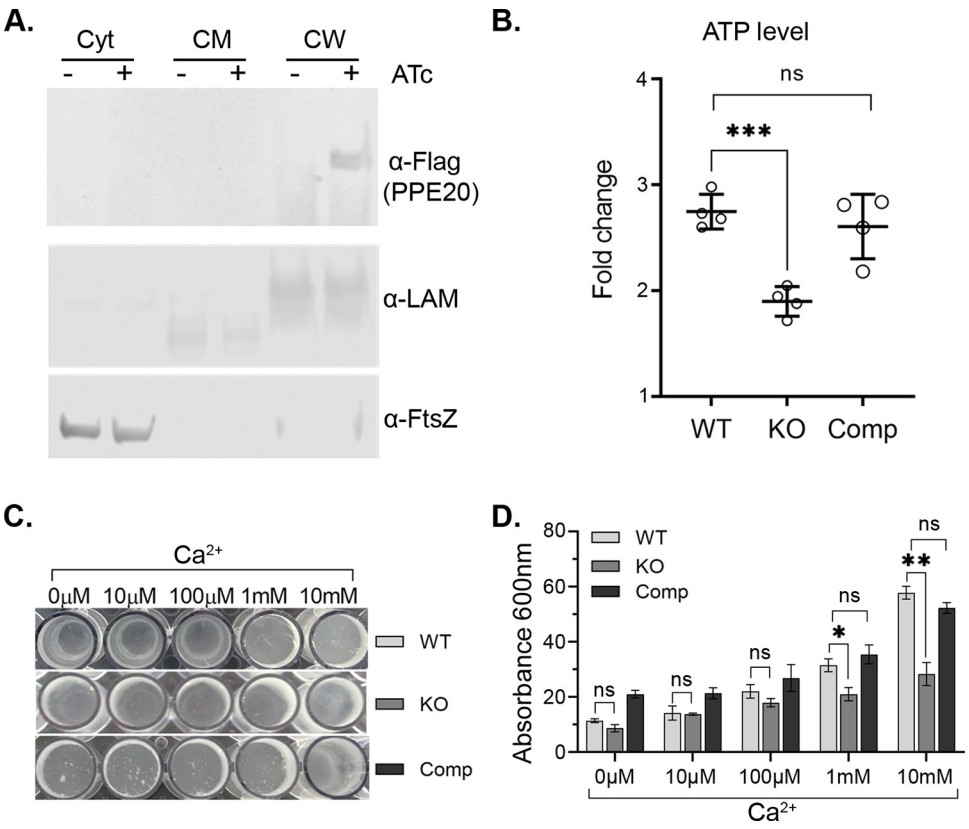

**Fig 3. *pe15/ppe20* affects $Ca^{2+}$ associated processes and facilitates $Ca^{2+}$ uptake.** (A) Western blot showing localization of PPE20 to the cell wall. Subcellular fractions from the complemented *pe15/ppe20* KO strain expressing PPE20-FLAG were probed with α-FLAG antibody. Cyt: cytosolic fraction, CM: cell membrane fraction, CW: cell wall fraction. LAM is a cell envelope marker and FtsZ is a cytosolic marker. (B) PE15/PPE20 affect cellular ATP levels. *Mtb* was treated with 1 mM $Ca^{2+}$ at 37°C and ATP quantified using BacTiter-Glo reagent. Fold change was calculated by comparing to the $Ca^{2+}$-free condition. Data shown are from 4 experiments, error bars show standard deviation ($^{ns}p > 0.05$, $^{***}p < 0.001$). (C) PE15/PPE20 affect biofilm formation. Cultures were grown with increasing concentrations of $Ca^{2+}$ and biofilm was quantified by crystal violet (D). The experiment was repeated 3 times, error bars show standard error ($^{ns}p > 0.05$, $^{*}p < 0.05$, $^{**}p < 0.01$). The data underlying all the plots and uncropped images in this figure are included in S1 Data and S1 Raw images. $Ca^{2+}$, calcium ion; Comp, complement; KO, knockout; WT, wild type.

tested the *pe15/ppe20* KO strain for altered ATP levels. The KO showed reduced $Ca^{2+}$-dependent increase in ATP production that was reverted to wild-type (WT) levels by complementation with *pe15/ppe20* (Comp) (Fig 3B). We next tested whether the PE15/PPE20 complex also affects $Ca^{2+}$-dependent biofilm formation. The KO blocked the effect of $Ca^{2+}$ on biofilm formation, and complementation with *pe15/ppe20* restored the phenotype (Fig 3C and 3D). These data show that the PE15/PPE20 complex is not only transcriptionally responsive to, but regulates functions related to $Ca^{2+}$.

## The PE15/PPE20 complex is a $Ca^{2+}$ importer

A recent study showed selective channel function of PE/PPE proteins [15], and the decrease of transcript levels of *pe15/ppe20* in the presence of $Ca^{2+}$ was consistent with the behavior of a $Ca^{2+}$ channel. To directly test the idea that PE15/PPE20 has a role in $Ca^{2+}$ transport, we created an *Mtb* reporter strain using a fluorescence resonance energy transfer (FRET) system that detects intracellular $Ca^{2+}$, Twitch [24]. Twitch is based on a minimal $Ca^{2+}$-binding domain from troponin C that is optimized for maximal $Ca^{2+}$ selectivity and ratiometric signal and that has a $K_D$ for $Ca^{2+}$ of 200 nM [24] (Fig 4A). We ectopically expressed Twitch in the H37Rv background (*Mtb-twitch*) and tested for Twitch expression in different cell fractions. Twitch was expressed only in the cytoplasm, indicating that it only reports on cytoplasmic $Ca^{2+}$ (S3A Fig). We could only obtain a robust FRET signal in $Ca^{2+}$- free Sauton's medium, not 7H9 medium which already contains $Ca^{2+}$ (Figs 4B, S1C and S1D). We thus used Sauton's medium for all $Ca^{2+}$ measurements. To further test the Twitch reporter in *Mtb*, we measured the $Ca^{2+}$ signal at different $Ca^{2+}$ concentrations. The FRET signal was robust, dose dependent, and comparable to that of Twitch in previously described nonbacterial systems [24] (Fig 4C). We next tested the selectivity of the probe over the closest Earth alkali metal neighbor, $Mg^{2+}$. $Mg^{2+}$ did not generate a FRET signal (S3B Fig). These data show that Twitch is a sensitive probe to continuously measure intracellular $Ca^{2+}$ in *Mtb*, establish conditions for detecting intracellular $Ca^{2+}$ changes, show that *Mtb* readily takes up extracellular $Ca^{2+}$, and reveal a similar *Mtb* response to extracellular $Ca^{2+}$ to that previously observed in *E. coli* [25].

To directly test for a role of the PE15/PPE20 complex in $Ca^{2+}$ transport, we measured $Ca^{2+}$ uptake in the *Mtb-twitch* strain and a strain expressing *Mtb-twitch* in the *pe15/ppe20* KO background. The WT produced a robust FRET signal as before upon addition of 5 mM $Ca^{2+}$. The KO, however, showed reduced influx, consistent with the loss of a selective import channel (Fig 4D). Complementation of the KO with *pe15/ppe20* expressed from an extrachromosomal plasmid restored $Ca^{2+}$ levels to those in WT. PDIM is one of the most abundant cell wall lipids and is often lost in strains grown in vitro [26]. To rule out PDIM differences between WT and mutant strains that might affect $Ca^{2+}$ transport, we tested all strains for PDIM. All strains produced PDIM at comparable levels (S3C Fig). To further test if PE15/PPE20's transport role is associated with the cell wall/outer membrane, we permeabilized the *Mtb* cell wall by treatment with lysozyme and Triton-X100 [27]. The difference in $Ca^{2+}$ influx between WT and KO was reduced by permeabilization, indicating that PE15/PPE20 facilitate transport across the outer membrane (S3D Fig).

## Discussion

$Ca^{2+}$ signaling is ubiquitous in eukaryotes [1] but in bacteria, only few components and functions of $Ca^{2+}$ signaling have been described. In *Mtb*, the outer membrane generally prevents passage of charged solutes, presenting a hurdle for $Ca^{2+}$ uptake not encountered in most bacteria. In the absence of typical porins, the transport processes that allow for transfer through the outer membrane have long been unknown. Here, we identify a specific $Ca^{2+}$ channel that

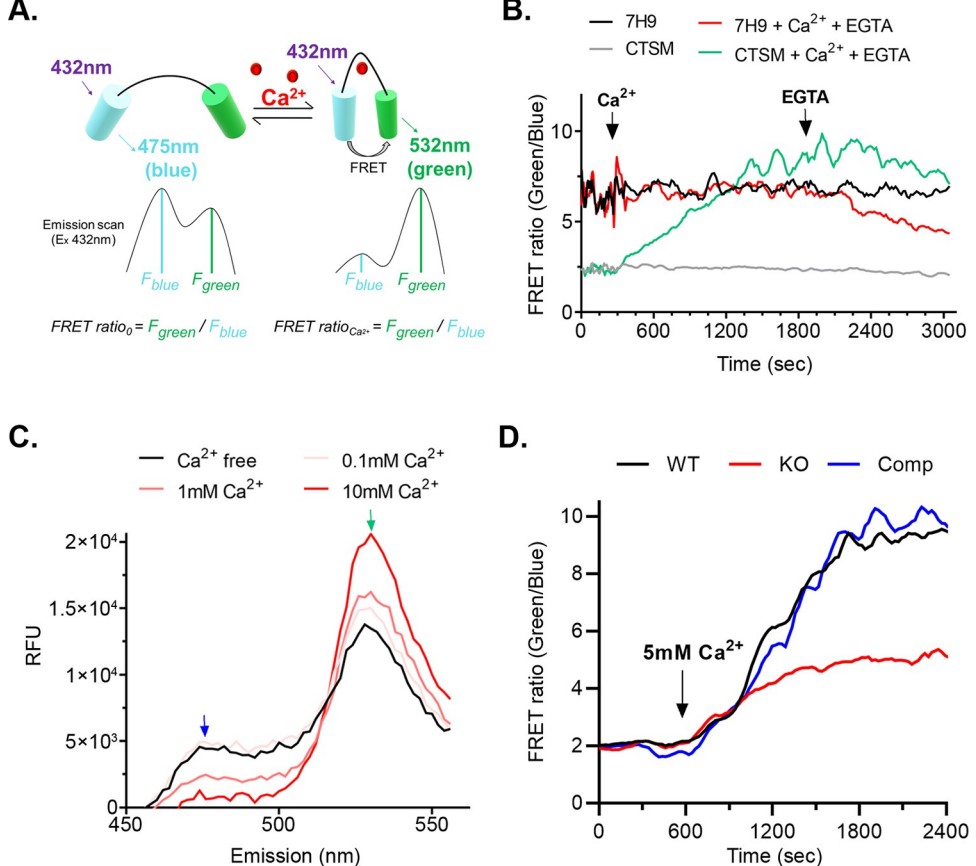

**Fig 4. PE15/PPE20 facilitate Ca$^{2+}$ uptake.** (A) Schematic of the Ca$^{2+}$ FRET probe and its ratiometric signal. (B) Ca$^{2+}$ FRET signals in different media show that Ca$^{2+}$ presents in standard 7H9 compromise Ca$^{2+}$ detection. FRET ratio was calculated by calculating the green:blue fluorescence ratio and was plotted against time. (C) The FRET probe generates a robust, dose dependent FRET signal. Emission scan of *Mtb-twitch* incubated with increasing concentrations of Ca$^{2+}$ for 30 min at 37˚C. (D) FRET trace over time shows reduced Ca$^{2+}$ influx in the *pe15/ppe20* KO strain. The data underlying all the plots in this figure are included in S1 Data. Ca$^{2+}$, calcium ion; FRET, fluorescence resonance energy transfer; KO, knockout; WT, wild type.

consists of PE15 and PPE20. In addition, we identify several phenotypes associated with elevated Ca$^{2+}$ in *Mtb*: Ca$^{2+}$ leads to an increase in cellular ATP concentrations, an effect that has also been reported in *E. coli* and was suggested to be a mechanism to sustain the increased activity of Ca$^{2+}$ ATPase efflux pumps required to reset Ca$^{2+}$ levels [16]. We also show a clear contribution of Ca$^{2+}$ to biofilm formation. The role of biofilms for *Mtb* pathogenesis and tuberculosis treatment has long been unclear, but recent evidence supports the presence of biofilms in infected lungs of nonhuman primates and human patients and a role in *Mtb* pathogenesis and drug susceptibility [28].

The PE/PPE proteins have long been the subject of much speculation. They are specific not only to mycobacteria but are predominantly found in pathogenic or slow-growing mycobacteria. Although they make up approximately 10% of *Mtb*'s genetic coding potential, their function has long remained unclear [14]. Their large number and sequence variation is reminiscent of variable surface proteins that serve as antigenic decoys in other pathogens. Consistent with this idea, the PE/PPE proteins have generally been linked to the cell wall, although mass spectrometry-based proteomic studies not always identify PE/PPE proteins in cell wall fractions. The difficulties to detect PE/PPE proteins by MS are likely due to a

combination of the relative lack of trypsin cleavage sites in the repetitive sequences and the challenges to assign repeat-derived peptides to individual PE/PPE proteins [29–31]. However, phylogenetic analyses showed that sequence variation does not arise through immunogenic pressure and does not support a role in providing antigenic variation [14]. While mycobacterial ESX secretion systems have been associated with nutrient import before [32,33], the responsible proteins and mechanisms were not known. A recent hallmark study showed channel-like function of PE/PPE complexes specifically transporting several carbon sources, $Mg^{2+}$, and phosphate across the outer mycobacterial cell membrane [15]. We now show a PE/PPE complex with such a channel-like function for $Ca^{2+}$. What's more, we show direct binding of PPE20 to the cargo, which is also more consistent with a channel than for example, a porin. The selectivity of PE15/PPE20 for $Ca^{2+}$ over even the closely related $Mg^{2+}$ further argues for a specific channel rather than a porin, which often show more indiscriminate transport [34]. In fact, another PE/PPE pair, PE20/PPE31 specifically transports $Mg^{2+}$ [15]. Interestingly, several other PE-PGRS proteins bind $Ca^{2+}$ [35,36], could also contribute to $Ca^{2+}$ import, and could explain the residual $Ca^{2+}$ import seen in the *pe15/ppe20* KO strain. A link of PE15/PPE20 to pathogenicity is highly plausible given the association of PE/PPE proteins with the outer mycobacterial cell membrane [37], the presence of PE/PPE proteins primarily in pathogenic mycobacteria, their association with the type VII secretion system, and previous studies directly implicating other PE/PPE proteins in virulence [14]. What's more, the source of $Ca^{2+}$ for the PE/PPE channel is likely host $Ca^{2+}$. In this way, *Mtb* may be able to eavesdrop on the many host cell signaling events that involve $Ca^{2+}$, for example phagocytosis, which is $Ca^{2+}$-dependent [38] and marks the beginning of *Mtb's* intracellular stage.

How do PE/PPE proteins facilitate transport? Crystal structures show that these proteins, unlike classical porins, fold into long alpha helices, with no structural information available for the long repetitive C-terminal sequences of the PPE proteins that are intrinsically disordered. A potential model for how the alpha helical sections of PE/PPE proteins may form pores was recently suggested by structures of the ESX type VII secretion system component EspB [39], which is a naturally occurring fusion of a PE and a PPE protein that forms a donut-shaped heptamer with a large central pore. Although the helices that potentially traverse the cell membrane did not have the requisite outward-facing hydrophobic residues, other models in which the PPE C-termini provide a hydrophobic sheath or association with other membrane-spanning proteins of the ESX complex are possibilities. In our studies, $Ca^{2+}$ binding led to an atypical decrease in thermal stability of PPE20. This change could indicate relaxation of the protein concomitant with channel opening. The precise nature of this unfolding event and how it impacts transport remains to be identified, as do the larger transport mechanics of the PE/PPE proteins and how the intrinsically disordered regions of the PPE proteins contribute to transport. Although our data show that PE15/PPE20 are necessary for efficient $Ca^{2+}$ uptake, they may not be sufficient. The genetic and functional association of the PE/PPE proteins with the ESX secretion system suggests that other ESX components such as an ATPase could be required for PE/PPE transport functions.

The PE and PPE proteins can be further stratified into subfamilies by C-terminal sequence motifs [14] that perhaps also indicate distinct functions. While all currently known PE/PPE channels including PE15 contain a minimal PE-only protein, the PPE proteins identified as components of transporters prior to our study belong to the PPE-SVP subfamily. However, PPE20 belongs to the PPE-PPW subgroup and is the first example of a PE/PPE channel outside of the PPE-SVP group. Similarly, the previous PE/PPE pairs with channel function are associated with the ESX-5 secretion system [40], while PE15/PPE20 is associated with ESX-3 [37]. The PE/PPE channels appear to be highly specific, which could explain their large number, as *Mtb* requires access to many different nutrients without compromising the outer membrane's

barrier function. The full set of PE/PPE transporters and cargos remains to be identified. However, our data support the emerging theme that the PE/PPE protein family is a mycobacterial transporter family that solves the conundrum of an outer cell membrane so impermeable as to exclude vital nutrients.

## Methods

### Media and growth conditions

*Mtb H37Rv* was used as a parental strain for generation of all mutants and was cultured at 37˚C in either Middlebrook 7H9 medium (Difco) with 10% (vol/vol) oleic acid-albumin-dextrose-catalase (OADC) enrichment (BBL; Becton Dickinson), 0.5% glycerol (referred to as "7H9+GO"), and 0.05% Tween 80 or 0.05% Tyloxapol (referred to as "7H9+GOT or 7H9+-GOTy") or 7H10 agar supplemented with 10% OADC and 0.5% glycerol or Chelex-treated Sauton's medium (CTSM) with additional supplements as indicated. CTSM consisted of 0.5 g $KH_2PO_4$, 4g L-asparagine, 2 g citric acid, 6% glycerol, adjusted to pH 7, and was treated overnight with Chelex-100 resin (10 g/L) (Sigma) to remove trace metal ion contaminants including $Ca^{2+}$. After filtration, 0.5 g/L of $MgSO_4$, 0.05 g/L ferric ammonium citrate, 0.1 ml of 1% zinc sulfate, and 0.05% Tween 80 or 0.05% Tyloxapol was added. The pH was adjusted to 6.9 and the medium was sterilized by filtration. Strains bearing antibiotic cassettes were cultured with 50 μg/mL hygromycin or 30 μg/mL kanamycin or 25 μg/ml zeocin as appropriate. $Ca^{2+}$ concentrations in media were determined using ICP-OES (Perkin Elmer Optima 8300). Media were treated with nitric acid (trace metal grade) for 1 h at 65˚C and diluted with Chelex-treated water. $Ca^{2+}$ intensities in each of the digested sample were measured 3 times by ICP-OES at 317.933 nm. $Ca^{2+}$ standards were measured to generate a standard curve of absorbance versus concentration (S1C Fig). Intensities were converted to $Ca^{2+}$ concentration using the standard curve.

### Biofilm formation

*Mtb* biofilms were generated and quantified according to a published protocol [41] in 48-well polystyrene plates. Briefly, cells were grown to an $OD_{600}$ of 0.8–1 in 7H9+GOT medium, washed twice with CTSM, and diluted to an $OD_{600}$ of 0.01 in CTSM without detergent and added to each well supplemented with varying concentrations of $Ca^{2+}$. Outer wells were filled with water and the plate was incubated at 37˚C for 4 weeks without shaking. The plates were photographed and the processed for crystal violet staining. The medium was removed, biofilms were dried and incubated with 500 μl of 1% crystal violet for 10 min. Wells were washed 3 times with water and dried again. Absolute ethanol (1 ml) was added to each well and incubated for 10 min. Then, 3-fold serial dilutions were read at $A_{600}$ on a spectrophotometer in a 96-well plate. The represented bar graph is the average readings of 4 biological replicates and Welch's *t* test was applied to determine the significance (*p*-value).

### ATP measurement

*Mtb* cells from a 7H9+GOT medium culture ($OD_{600} \sim 1$) were washed twice and diluted to an $OD_{600}$ of 0.01 in CTSM without detergent supplemented with different concentrations of $CaCl_2$ or 1 mM EGTA. Cultures were grown in a 48-well plate at 37˚C for 14 days. Cells were treated with 0.1% Tween-80 overnight to create a homogenous suspension. ATP was quantified by incubating 50 μl of the bacteria in triplicate with 50 μl of the BacTiter-Glo reagent for 15 min followed by a luminescence reading. ATP production in the $Ca^{2+}$-treated cultures were calculated as fold change compared to the calcium-free condition. The represented bar graph

is the average fold change of 4 biological replicates and Welch's *t* test was applied to determine the significance (*p*-value).

## RNA sequencing

*Mtb* H37Rv was grown to an $OD_{600}$ of approximately 1 in 7H9+GOT medium. Cells were washed twice with CTSM, subcultured in CTSM starting at an $OD_{600}$ of 0.05, and grown to an $OD_{600}$ of approximately 1. The culture was again subcultured in CTSM starting at an $OD_{600}$ of 0.05 and incubated at 37°C to an $OD_{600}$ of 0.2. The cultures were supplemented with or without 1 mM $CaCl_2$ in triplicate. At the indicated times, cells were pelleted at 4,000 g for 5 min at 4°C, resuspended in Trizol and lysed by bead beating for 30 s at 6 m/s for 3 cycles with intermittent cooling on ice. Cell debris were pelleted at 20,000 g for 1 min, and the supernatant was transferred to a heavy phase lock gel tube containing 300 μl chloroform. The tubes were inverted for 2 min and centrifuged at 20,000 g for 5 min. RNA in the aqueous phase was precipitated using 300 μl isopropanol and 300 μl high salt solution (0.8 M Na citrate, 1.2 M NaCl). RNA was purified using QIAGEN RNeasy kit and ribosomal RNA was depleted using the Ribo-Zero rRNA removal magnetic kit (Illumina). The cDNA library was generated using the NEBNext Ultra II RNA Library Prep Kit, and each replicate was barcoded in the DNA library using the NEBNext Multiplex Oligos for Illumina. Libraries were quantified using the KAPA qPCR quantification kit, pooled, and sequenced at the University of Washington Northwest Genomics Center with the Illumina NextSeq 500 High Output v2 Kit. Read alignment was performed using the Bowtie 2 custom processing pipeline (https://github.com/robertdouglasmorrison/DuffyNGS, https://github.com/robertdouglasmorrison/DuffyTools). Gene expression changes were identified using a combination of 5 differential expression (DE) tools within DuffyTools. The 5 DE tools included round robin, RankProduct, significance of microarrays (SAMs), EdgeR, and DeSeq2. Each DE tool measurement was combined using the weighted average of fold change and significance (*p*-value). Genes with averaged absolute fold change more than 1.5-fold and *p*-value <0.01 were considered differentially expressed. RNA-seq data is available at NCBI-GEO (accession no. GSE214266).

## Cloning, co-expression, and purification of recombinant PE15 and PPE20 proteins

pETDuet dual expression plasmid was used to co-express PE15 and PPE20. PE15 was cloned in the MCS-1 region with N-terminal His tag and PPE20 was cloned in the MCS-2 region with a C-terminal Strep II tag. *Mtb* Rv1386 (PE15) and Rv1387 (PPE20) genes were amplified from *Mtb* H37Rv genomic DNA using the primers Duet 1–5 that included Gibson overlap sequence (Table B in S1 Table). The linearized pETDuet plasmid and the region between MCS-1 and MCS-2 were prepared by PCR amplification using the primers Duet 6,7 and Duet 8,9, respectively (Table B in S1 Table). Finally, all 4 purified PCR products, i.e., PE15, PPE20, region between MCS-1/MCS-2, and linearized pETDuet plasmid were ligated using the Gibson Assembly (NEB) to generate the pETDuet pe15/ppe20 plasmid. Cloning was confirmed by sequencing. The vector was transformed into *E. coli BL21(DE3)* and a single colony was picked and was grown at 37°C in Terrific Broth medium containing ampicillin. Protein expression was induced by addition of 0.5 mM isopropyl-β-d-thiogalactoside (IPTG) at $OD_{600}$ of approximately 0.4; culture was then maintained at 16°C for 16 h. Cells were harvested, and pellets were processed for purification. For complex purification through His-PE15, pellets were resuspended in buffer A (50 mM Tris (pH 8.0), 150 NaCl, and 10% glycerol) containing 20 mM imidazole and 1 mM AEBSF and lysed by sonication. Lysate was centrifuged at 35,000 g for 30 min at 4°C and the supernatant was loaded on Ni-NTA column. The column was

thoroughly washed with buffer A containing 20 mM imidazole followed by elution with buffer A containing 250 mM imidazole. For PPE20-strep II tag, pellets were resuspended in buffer A containing 1 mM AEBSF and lysed by sonication. Lysate was centrifuged at 35,000 g for 30 min at 4˚C and the supernatant was loaded on Strep-tactin column. The column was thoroughly washed with buffer A followed by elution with buffer A containing 2.5 mM desthiobiotin. Both the purified proteins were immediately dialyzed against buffer A and stored at −80˚C. Interaction of PE15/PPE20 was confirmed by western blot. The wash and elution fraction from the Ni-NTA and strep tag purification were loaded on either SDS-PAGE or Native-PAGE, and the proteins were transferred onto a nitrocellulose membrane. Blots were probed with mouse α-His antibody followed by IRDye 680RD Goat α-Mouse IgG Secondary Antibody (LI-COR) to detect the His tagged PE15 and were also probed with IRDye 800CW Streptavidin to detect the PPE20-strep tag. The blots were scanned on LI-COR Odyssey platform using the 700 nm and 800 nm channel.

## Thermal shift assay and equilibrium dialysis to determine Ca²⁺ binding

Purified PE15/PPE20 protein was incubated with or without 1 mM $CaCl_2$ in a thermocycler for 5 min at 25˚C followed by incubation at varying temperature (55 to 70˚C) for 3 min. The reaction was cooled to 4˚C and centrifuged at 20,000 g for 30 min at 4˚C to separate the native (supernatant) and the denatured (pellet) fractions. Pellet fraction was solubilized by boiling with 2xSDS loading buffer. PPE20 in each of the supernatant and pellet fraction was detected by western blot using rabbit α-strep II tag antibody followed by IRDye 800CW Goat α-Rabbit IgG Secondary Antibody. To determine the $K_D$ value, PPE20 was incubated with varying concentration of $CaCl_2$ (12.5 μm to 3.2 mM) for 5 min at 25˚C, 3 min at 65˚C followed by incubation at 4˚C. PPE20 in the pellet fraction was detected by western blot as described above. The band intensities were calculated using the Image studio software and converted to fraction denatured relative to the total protein. The graph of fraction denatured v/s log $Ca^{2+}$ concentration was plotted using GraphPad Prism 9.3, and the $K_D$ value was determined by fitting a dose-response curve using the least square regression method. For equilibrium dialysis, $Ca^{2+}$-free PE15/PPE20 was prepared by EGTA treatment and dialysis against Chelex-treated buffer A. Protein (10 μm) was placed in the upper chamber and dialyzed against 50 mM Tris (pH 7.4), 150 mM NaCl buffer containing 200 μm of $Ca^{2+}$ in the lower chamber for 24 h and the $Ca^{2+}$ concentration in both chambers was determined by ICP-OES. Samples (100 μl) were digested with 100 μl of nitric acid (trace metal grade) for 1 h at 65˚C and diluted with 2.8 ml of Chelex-treated water. $Ca^{2+}$ intensities were measured 3 times by ICP-OES and converted to $Ca^{2+}$ concentration using the standard curve described above. Percentage ΔC was calculated as per the following formula:

$$\%\Delta C = \frac{Ca^{2+}(upper\ chamber) - Ca^{2+}(lower\ chamber)}{Ca^{2+}(lower\ chamber)} \times 100.$$

The thermal shift and equilibrium dialysis experiments were repeated with buffer control (no protein), calcium-binding protein (CBP) from *Encephalitozoon cuniculi* as a positive control (obtained from SSGCID # EncuA.01276.a.A1.PS00484), and Rv0831c as a negative control. The bar graph in Fig 2G is the average %ΔC of 3 technical replicates. Student's *t* test was applied to determine the *p*-value.

## Creation of *pe15/ppe20* knock out and complemented strain in *Mtb* H37Rv

The KO strain was created by recombineering as described previously [23]. Initially 500 bp upstream of *pe15* (primers KO1 and KO2), 500 bp downstream of *ppe20* (primers KO3 and

KO4) and hygromycin cassette (primers KO5 and KO6) were amplified separately (Table B in S1 Table). Both PCR fragments were Gibson ligated to the 5′ and 3′ of a hygromycin-resistance cassette, respectively, to generate the recombineering cassette. This linear recombineering cassette was PCR amplified, purified, and electroporated into *Mtb* H37Rv strain carrying recombineering plasmid pNIT [42]. Hygromycin positive colonies were screened, and the positive clone was confirmed by DNA sequencing. For the complemented strain, both *pe15* (His-tag) and *ppe20* (FLAG-tag) genes were separately PCR-amplified using primers Comp 1–3 and Comp 4, 5, respectively (Table B in S1 Table), and Gibson cloned in pDTCF plasmid (Zeocin) separately under an anhydrotetracycline (ATc)-inducible promoter. The ATc promoter along with the *ppe20* gene was amplified using primers Comp 6, 7 (Table B in S1 Table) from pDTCF-PPE20 vector and Gibson cloned in pDTCF-PE15 vector downstream of *pe15* gene. The resulting plasmid pDTCF-*pe15/ppe20* carries both genes with individual ATc promoters. The plasmid was electroporated into the KO strain and positive clones were selected by growth in hygromycin and zeocin. This strain expressed His-PE15 and PPE20-FLAG proteins when induced with 100 ng/ml ATc.

### Creation of *Mtb* Ca$^{2+}$ FRET reporter strain

The Ca$^{2+}$ FRET reporter Twitch-2B was PCR amplified from the plasmid Twitch-2B pRSETB obtained from Addgene [24] using the primers T2B1 and T2B2 (Table B in S1 Table) that included Gibson overlap sequence. The PCR product was Gibson cloned in the *E. coli-mycobacterial* episomal shuttle vector pDTCF (kanamycin) under the anhydrotetracycline (ATc)-inducible promoter. The plasmid was electroporated into wild type *Mtb* H37Rv (WT), *pe15/ppe20* KO and *pe15/ppe20* complemented strain (Comp), and the expression of Twitch was induced using 100 nM ATc. The expression of Twitch was confirmed by measuring the fluorescence of uninduced and induced cells using a plate reader at 3 different wavelengths (432/475 nm, 432/532 nm, and 488/532 nm). The localization of Twitch and PPE20 was confirmed by western blot. Briefly, 100 ml culture of uninduced and induced *Mtb-twitch* complemented with the *pe15/ppe20* genes was grown in 7H9+GO medium. Cells were washed twice in PBS and resuspended in lysis buffer (50 mM Tris (pH 7.4), 150 mM NaCl) containing protease inhibitor cocktail. Cells were lysed by bead beating and the following fractions were prepared according to published protocol [43]: cytosolic (CYT), cell membrane (CM), and cell wall (CW) fraction. Briefly, the lysate was sequentially centrifuged at 3,000 g to separate the beads and cell debris, followed by spin at 20,000 g to pellet the CW fraction and finally at 100,000 g to separate the pellet, i.e., CM fraction and supernatant CYT fraction. CM and CW fractions were washed once and resuspended in lysis buffer. The expression of Twitch protein (His-tagged) was detected by western blot using a mouse α-His antibody, PPE20 was detected by mouse α-Flag antibody and the purity of cellular fractions was confirmed by using α-LAM (CM and CW fraction) and α-FtsZ (CYT) antibodies (obtained from BEI resources, NIAID, NIH).

### PDIM analysis

Cultures were grown in 50 ml 7H9+GO for 7 days and washed 3× with PBS. Cell pellets were resuspended in 9 ml chloroform-methanol (2:1) and incubated overnight at room temperature with shaking. The extract was filtered through 0.2 μm PTFE filters and dried under a stream of nitrogen. The total lipid was weighed and resuspend at a concentration of 20 mg/ml in chloroform-methanol (2:1). Lipids (500 μg) were spotted onto aluminum-backed TLC plate and run in petroleum ether-ethyl acetate (98:2) as the solvent system. Control PDIM was obtained from BEI resources, solubilized in chloroform-methanol (2:1), and 100 μg was run on TLC.

The TLC plate was dried, and the lipids were stained by spraying 5% sulfuric acid in methanol followed by heating.

## $Ca^{2+}$ uptake assay

*Mtb-twitch* was grown in either 7H9+GOT or CTSM to an $OD_{600}$ of 1. Cells (50 μl) were aliquoted in duplicate wells and spiked with $Ca^{2+}$ (5 mM) and EGTA (5 mM) at indicated time points and fluorescence was continuously measured at wavelengths 432/475 nm (blue) and 432/532 nm (green) at 37°C. Fluorescence readings were corrected by subtracting the readings from wells containing uninduced culture. The FRET ratio was calculated as the green:blue fluorescence ratio and plotted against time. An increase in the FRET ratio indicates an increased intracellular $Ca^{2+}$ concentration. To determine the dose-dependent effect of $Ca^{2+}$ on the FRET signal, *Mtb-twitch* was incubated with increasing concentration of $Ca^{2+}$ for 30 min at 37°C and an emission scan (from Em450-550 nm) was recorded using excitation at 432 nm.

*Mtb* H37Rv (WT), *pe15/ppe20* KO and complemented strains (Comp) expressing *twitch* were grown to $OD_{600}$ of approximately 0.5 in CTSM/tyloxapol with or without 100 nM ATc. Cells were washed twice and resuspended in CTSM/tyloxapol at a dilution of 1 OD/ml. Cells were spiked with $Ca^{2+}$ (5 mM) at indicated time point and blue and green fluorescence were measured continuously at 37°C. FRET ratios were calculated and plotted against time. The assay was repeated by treating cells with permeabilizing reagent [27] (0.1% Triton-X100 and 2 μg/μl lysozyme) at time 0 min. Experiments were performed using 3 biological replicates and 2 technical replicates.

## Supporting information

**S1 Fig. Growth curves of Mtb H37Rv and $Ca^{2+}$ quantification in different media.** (A) CTSM and (B) 7H9 medium containing glycerol, OADC, and Tween-80 (7H9+GOT) supplemented with different concentrations of $CaCl_2$ shows no effect of $Ca^{2+}$ on growth. (C) ICP-OES standard curve for the determination of $Ca^{2+}$ concentrations. (D) $Ca^{2+}$ concentrations in standard 7H9 and Sauton's media. The data underlying all the plots in this figure are included in S1 Data. $Ca^{2+}$, calcium ion; CTSM, Chelex-treated Sauton's medium; ICP-OES, inductively coupled plasma optical emission spectrometry.
(TIF)

**S2 Fig. Thermal shift-based analysis of $Ca^{2+}$ binding to control proteins.** (A) A known CBP from *Encephalitozoon cuniculi* shows a $Ca^{2+}$-dependent increase in the melting temperature and concomitant precipitation. (B) A protein not known to bind $Ca^{2+}$ shows no difference in melting temperature and resulting precipitation. Uncropped images in this figure are included in S1 Raw images. $Ca^{2+}$, calcium ion; CBP, calcium-binding protein.
(TIF)

**S3 Fig. Transport by PE15/PPE20 is associated with the cell wall and is specific for $Ca^{2+}$.** (A) Western blot showing localization of Twitch-2B (His tagged) in the cytosolic fraction. The same cell fractions as in Fig 3A were used, see controls therein. Cyt: cytosolic fraction, CM: cell membrane fraction, CW: cell wall fraction. (B) The FRET signal is highly specific for $Ca^{2+}$ over the congener $Mg^{2+}$. The His-tagged Twitch protein was purified from the cytosolic fraction using Ni-NTA column, treated with EDTA and desalted. An emission scan of the purified protein was recorded upon incubation with either $Ca^{2+}$ or $Mg^{2+}$. (C) Thin-layer chromatography of the cell wall component PDIM shows no difference in PDIM content between WT and *pe15/ppe20* KO strains. (D) Permeabilization of the cell wall diminishes PE15/PPE20's effect on $Ca^{2+}$ import. WT, *pe15/ppe20* KO, and complemented strain were transformed with

Twitch, permeabilized (P) with lysozyme and Triton-X100, and the $Ca^{2+}$-FRET signal measured. The data underlying all the plots and uncropped images in this figure are included in S1 Data and S1 Raw images. $Ca^{2+}$, calcium ion; FRET, fluorescence resonance energy transfer; KO, knockout; PDIM, phthiocerol dimycocerosate; WT, wild type.
(TIF)

**S1 Table. List of differentially expressed genes and primers.** (A) Differentially expressed genes in response to $Ca^{2+}$ in *Mtb* and (B) primers used in this study.
(PDF)

**S1 Data. Numerical data points underlying presented graphs.**
(XLSX)

**S1 Raw images. Uncropped western blot images.**
(PDF)

## Acknowledgments

Recombinant calcium-binding protein from *Encephalitozoon cuniculi* was provided by the Seattle Structural Genomics Center for Infectious Disease (www.SSGCID.org) that is supported Federal Contract No. HHSN272201700059C from the National Institute of Allergy and Infectious Diseases, National Institutes of Health, Department of Health and Human Services.

## Author Contributions

**Conceptualization:** Vishant Boradia, Christoph Grundner.

**Data curation:** Vishant Boradia.

**Formal analysis:** Vishant Boradia, Christoph Grundner.

**Funding acquisition:** Christoph Grundner.

**Investigation:** Vishant Boradia, Andrew Frando.

**Methodology:** Vishant Boradia.

**Project administration:** Christoph Grundner.

**Resources:** Christoph Grundner.

**Supervision:** Christoph Grundner.

**Visualization:** Vishant Boradia.

**Writing – original draft:** Vishant Boradia, Christoph Grundner.

**Writing – review & editing:** Vishant Boradia, Christoph Grundner.

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
