## [Editor Report · Decision Letter 0]

24 Oct 2022

Dear Dr. Grundner, 

Thank you for submitting your manuscript entitled "Calcium transport by the Mycobacterium tuberculosis PE15/PPE20 proteins" for consideration as a Research Article by PLOS Biology.

Your manuscript has now been evaluated by the PLOS Biology editorial staff, as well as by an academic editor with relevant expertise, and I am writing to let you know that we would like to proceed towards publication of your manuscript as a *Discovery Report*. For this format, you will need to reduce the number of figures from 5 to 4 and we won't require further peer-review. 

Discovery Reports describe novel and intriguing initial findings with the potential to lead to a significant new result for the field. Discovery Reports are short articles, typically with 2-4 main figures. While the research may be preliminary, studies should be advanced to the stage where observations or findings have been confirmed by independent methods or experimental approaches and obvious alternative interpretations have been ruled out. Discovery Reports are designed to work together with Update Articles to empower researchers to evaluate and share work in a way that more closely mirrors the real-world research process and create a comprehensive research story. 

However, before we can proceed, we need you to complete your submission by providing the metadata that is required. To this end, please login to Editorial Manager where you will find the paper in the 'Submissions Needing Revisions' folder on your homepage. Please click 'Revise Submission' from the Action Links and complete all additional questions in the submission questionnaire.

Once your full submission is complete, your paper will undergo a series of checks. After your manuscript has passed the checks we can proceed to the next step. To provide the metadata for your submission, please Login to Editorial Manager (https://www.editorialmanager.com/pbiology) within two working days, i.e. by Oct 26 2022 11:59PM.

Kind regards,

Paula

---

Senior Editor

PLOS Biology

---

## [Editor Report · Decision Letter 1]

28 Oct 2022

Dear Dr. Grundner,

Thank you for your patience while your manuscript "Calcium transport by the Mycobacterium tuberculosis PE15/PPE20 proteins" was being assessed at PLOS Biology. It has now been evaluated by the PLOS Biology editors and an Academic Editor with relevant expertise. 

Based on the reviews from Review Commons and on our Academic Editor's assessment of your revision, we are likely to accept this manuscript for publication, provided you satisfactorily address the following data and other policy-related requests.

1. DATA POLICY:

Please ensure that figure legends in your manuscript include information on where the underlying data can be found, and ensure your supplemental data file/s has a legend.

2. Please provide a blurb which (if accepted) will be included in our weekly and monthly Electronic Table of Contents, sent out to readers of PLOS Biology, and may be used to promote your article in social media. The blurb should be about 30-40 words long and is subject to editorial changes. It should, without exaggeration, entice people to read your manuscript. It should not be redundant with the title and should not contain acronyms or abbreviations.

3. We suggest a change in the title: "The Mycobacterium tuberculosis PE15/PPE20 complex transports calcium across the outer membrane".

We expect to receive your revised manuscript within two weeks. 

*Published Peer Review History*

*Press*

Sincerely,

Paula

---

Senior Editor,

pjaureguionieva@plos.org,

PLOS Biology

---

## [Editor Report · Decision Letter 2]

4 Nov 2022

Dear Dr. Grundner,

Thank you for the submission of your revised Discovery Report "The Mycobacterium tuberculosis PE15/PPE20 complex transports calcium across the outer membrane" for publication in PLOS Biology. On behalf of my colleagues and the Academic Editor, Matthew Waldor, I am pleased to say that we can in principle accept your manuscript for publication, provided you address any remaining formatting and reporting issues. These will be detailed in an email you should receive within 2-3 business days from our colleagues in the journal operations team; no action is required from you until then. Please note that we will not be able to formally accept your manuscript and schedule it for publication until you have completed any requested changes.

PRESS

Sincerely, 

Paula

---

Senior Editor

PLOS Biology
